# MASKEDFUSION360: RECONSTRUCT LIDAR DATA BY QUERYING CAMERA FEATURES

**Royden Wagner, Marvin Klemp, and Carlos Fernandez Lopez**
Karlsruhe Institute of Technology (KIT)
{firstname.lastname}@kit.edu

## ABSTRACT

In self-driving applications, LiDAR data provides accurate information about distances in 3D but lacks the semantic richness of camera data. Therefore, state-of-the-art methods for perception in urban scenes fuse data from both sensor types. In this work, we introduce a novel self-supervised method to fuse LiDAR and camera data for self-driving applications. We build upon masked autoencoders (MAEs) and train deep learning models to reconstruct masked LiDAR data from fused LiDAR and camera features. In contrast to related methods that use birds-eye-view representations, we fuse features from dense spherical LiDAR projections and features from fish-eye camera crops with a similar field of view. Therefore, we reduce the learned spatial transformations to moderate perspective transformations and do not require additional modules to generate dense LiDAR representations. Code is available at: https://github.com/KIT-MRT/masked-fusion-360

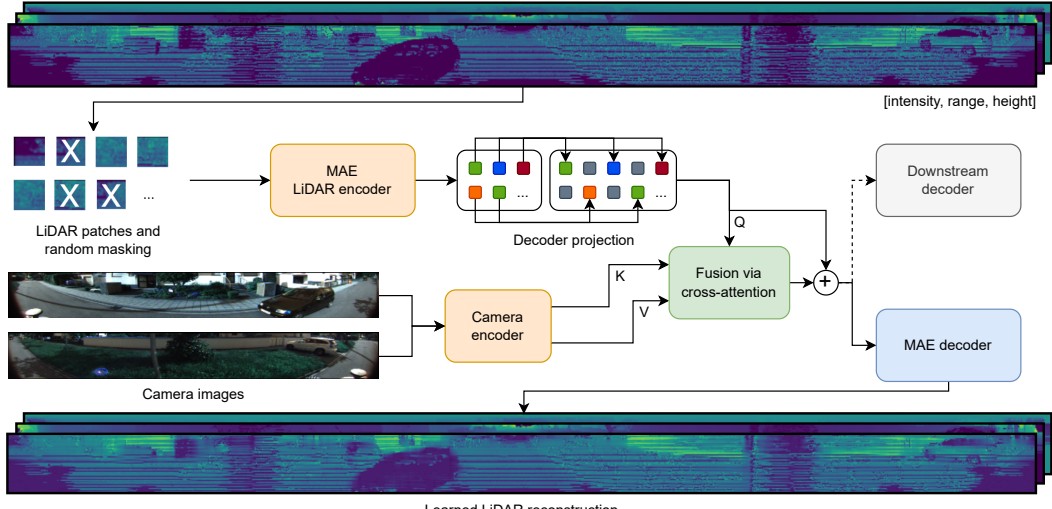

Figure 1: **Reconstruct LiDAR data by querying camera features.** Spherical projections of LiDAR data are transformed into patches, afterwards, randomly selected patches are removed and a MAE encoder is applied to the unmasked patches. The encoder output tokens are fused with camera features via cross-attention. Finally, a MAE decoder reconstructs the spherical LiDAR projections.

## 1 INTRODUCTION

Simple algorithms that scale well (He et al., 2016; Vaswani et al., 2017; Brown et al., 2020) drive deep learning research forward. A recent addition to this trend are self-supervised pre-training methods that leverage un-labeled data to improve downstream tasks in computer vision (He et al., 2020; Chen et al., 2020; Caron et al., 2021). For self-driving applications, self-supervised pre-training of perception models (Ma et al., 2019) or algorithms for domain adaptation (Wang et al., 2020) are

examples. LiDAR data provides accurate information about distances in 3D but lacks the semantic richness of camera data. Therefore, state-of-the-art methods for perception in urban scenes (Li et al., 2022; Piergiovanni et al., 2021) fuse data from both sensor types. In this work, we introduce a novel self-supervised method to fuse LiDAR and camera features for self-driving applications. We build upon masked autoencoders (MAEs) (He et al., 2022) and train vision transformers (ViTs) (Dosovitskiy et al., 2020) to reconstruct masked LiDAR data from fused LiDAR and camera features. Related methods (Ku et al., 2018; Li et al., 2022; Bai et al., 2022) fuse LiDAR and camera data in birds-eye-view (BEV) representations, which involves learning complex transformations from camera coordinates to birds-eye-view coordinates. Hess et al. (2023) and Min et al. (2022) apply masked autoencoding to voxel representations of LiDAR data, which requires additional modules for learning dense voxel embeddings from sparse LiDAR voxels. In contrast, we fuse features from dense spherical LiDAR projections with features from fish-eye camera crops with a similar field of view. Therefore, we reduce the learned spatial transformations to moderate perspective transformations. Moreover, our method does not require additional modules to generate dense LiDAR representations.

## 2 METHOD

Our proposed fusion method builds upon masked autoencoding (He et al., 2022), which is a recent form of denoising autoencoding. Masked autoencoders (MAEs) consist of a ViT-based encoder and a ViT-based decoder. Accordingly, input images are divided into patches and processed as a sequence of tokens. During training, 50% of these patches are randomly masked and the training target is the reconstruction of the masked patches. We use a MAE encoder as LiDAR encoder (Figure 1). After transforming the LiDAR data into patches, randomly selected patches are removed and the MAE encoder is only applied to the remaining patches. Therefore, our LiDAR encoder achieves the same computational efficiency as a vanilla MAE encoder. The encoder output tokens are projected to the decoder tokens by re-introducing tokens for masked patches. The decoder tokens are fused with camera features via cross-attention. For this attention mechanism, the LiDAR tokens serve as queries $Q$ and the camera tokens as keys $K$ and values $V$ (Figure 1, Appendix A). Finally, a MAE decoder reconstructs the spherical LiDAR projections from the fused tokens.

**Experiment.** We perform an initial evaluation of our method with the KITTI-360 dataset (Liao et al., 2022) (Appendix B). Figure 2 shows crops of a masked LiDAR intensity channel and the corresponding reconstruction and target. The importance of camera features is demonstrated by the superior reconstruction quality (higher MSSIM scores (Wang et al., 2003)) when LiDAR and camera features are used. We simulate missing camera features by using zero matrices as camera input.

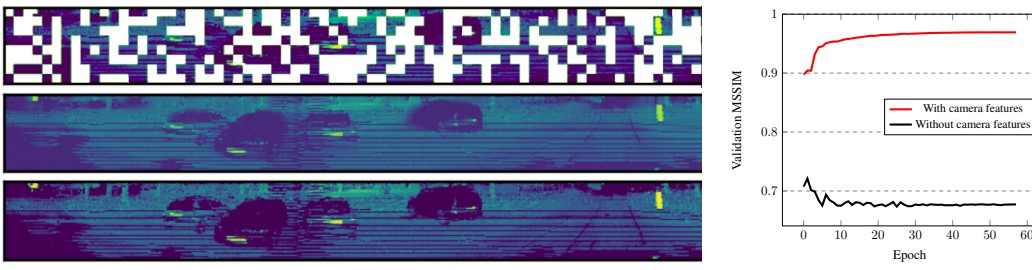

Figure 2: **Left:** Masked spherical LiDAR projection and corresponding reconstruction and target. **Right:** Comparison of reconstruction quality with camera features vs. without camera features.

## 3 CONCLUSION

In contrast to related methods that use birds-eye-view representations, we fuse features from dense spherical LiDAR projections and features from fish-eye camera crops. Therefore, we reduce the learned spatial transformations to moderate perspective transformations and do not require additional modules to generate dense LiDAR representations. Future steps include evaluating the performance of our fusion method as pre-training for semantic scene understanding in urban scenarios.

ACKNOWLEDGEMENTS

This work was accomplished within the project HAIBrid (FKZ 01IS21096A). We acknowledge the financial support for the project by the Federal Ministry of Education and Research of Germany (BMBF). Furthermore, this work was supported by the Helmholtz Association's Initiative and Networking Fund on the HAICORE@FZJ partition.

URM STATEMENT

The authors acknowledge that at least one key author of this work meets the URM criteria of ICLR 2023 Tiny Papers Track.

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

## A  FUSION VIA CROSS-ATTENTION

The ViT-based encoders for LiDAR and camera data process their inputs as sequences of patch tokens ([Patch]) and a learnable class token ([CLS]). We use the class token of the LiDAR encoder and the patch tokens of the camera encoder to fuse information via cross-attention (Chen et al., 2021). As shown in Figure 3, the LiDAR class token is used as queries vector and concatenated with the camera patch tokens to generate keys and values matrices for a standard attention module. Afterwards, the LiDAR class token is added to the attention output to compute a fused class token. In this way, the additional LiDAR class token can learn where LiDAR data is sparse or masked and query the camera tokens accordingly. Furthermore, this cross-attention mechanism is computed in a token-to-sequence manner. This reduces the computational complexity compared to the vanilla sequence-to-sequence manner from $O(n^2)$ to $O(n)$, where $n$ is the sequence length.

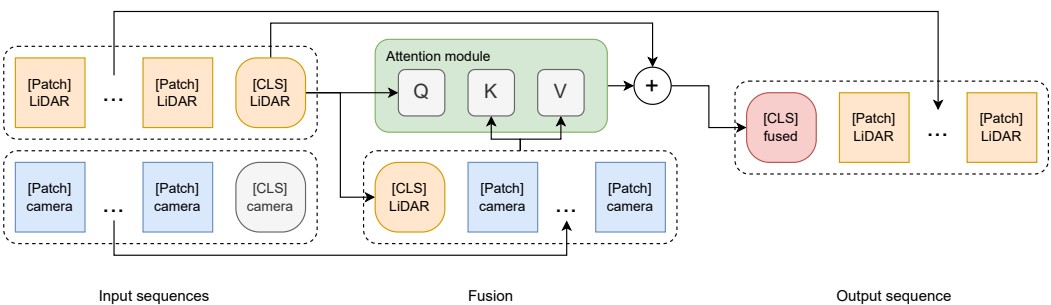

Figure 3: **Fuse LiDAR and camera tokens via cross-attention.** The attention module computes: softmax$(\frac{QK^T}{\sqrt{d_K}}) \cdot V$, where $Q$, $K$, and $V$ are query, key, and value vectors, and $d_K$ is the size of key vectors.

## B  EXPERIMENTAL SETUP AND FURTHER RESULTS

**Dataset.** KITTI-360 contains recordings from driving in urban and sub-urban areas in Germany. The dataset contains 76k sets of fish-eye camera images and LiDAR scans as training data and 13k sets as testing data. We use randomly selected 70k and 6k samples from the training data as training and validation splits and the testing data as test split. Following Meyer et al. (2019), we transform all LiDAR scans into spherical projections and store intensity, range, and height data as 3 channels (Figure 4).

**Model and training.** We use a patch size of $8 \times 8$ pixels for our model. The smaller patch size than in a vanilla ViT ($16 \times 16$) is chosen to better handle fine granular semantic correlations (Xie et al., 2021). Both our ViT-based encoders have an embedding dimension of 2048 and a depth of 8. The cross-attention block for fusion has an embedding dimensions of 1024 and a depth of 2. The reconstruction decoder has an embedding dimensions of 1024 and a depth of 8. As loss, we compute the mean squared error between the target and the reconstructed patches. We choose AdamW (Loshchilov & Hutter, 2019) as optimizer with an initial learning rate of $10^{-4}$ and use cosine annealing (Loshchilov & Hutter, 2017) to reduce the learning rate while training for 57 epochs. During training, we track the multiscale structural similarity index measure (MSSIM) between the learned reconstruction and the training target to asses the reconstruction quality.

**Results.** Table 1 shows the achieved MSSIM scores after training for 57 epochs. For all dataset splits, the reconstruction quality with camera features is significantly higher (30% higher MSSIM).

|  | MSSIM$_{\text{train}}$ | MSSIM$_{\text{val}}$ | MSSIM$_{\text{test}}$ |
|---|---|---|---|
| With camera features | 0.9694 | 0.9691 | 0.9612 |
| Without camera features | 0.6772 | 0.6771 | 0.6410 |

Table 1: Reconstruction quality for different dataset splits

**Comparison to masked autoencoding without camera features.** We evaluate the reconstruction performance of our model with and without camera features. The evaluation mode without camera features is similar to vanilla masked autoencoding but with spherical projections of LiDAR data instead of natural images as inputs and targets. In this mode, the camera features are zero matrices and the corresponding camera patch tokens are zero vectors. Therefore, in the cross-attention mechanism for information fusion (Figure 3), the LiDAR class token only attends to itself and is forwarded via the skip connection. Hence, our model becomes a masked autoencoder with only one encoder. As shown in Figure 2 on the right, on average 30% higher MSSIM scores are achieved with camera features vs. without camera features. Furthermore, the reconstruction quality without camera features decreases during training (0.72 MSSIM at epoch 2 vs. 0.68 MSSIM at epoch 57). This shows that our model learns to leverage camera features during training.

**Qualitative results.** Figure 4 shows additional qualitative results of our method. Overall, after training, our model is able to reconstruct the spherical LiDAR projections using our fusion algorithm.

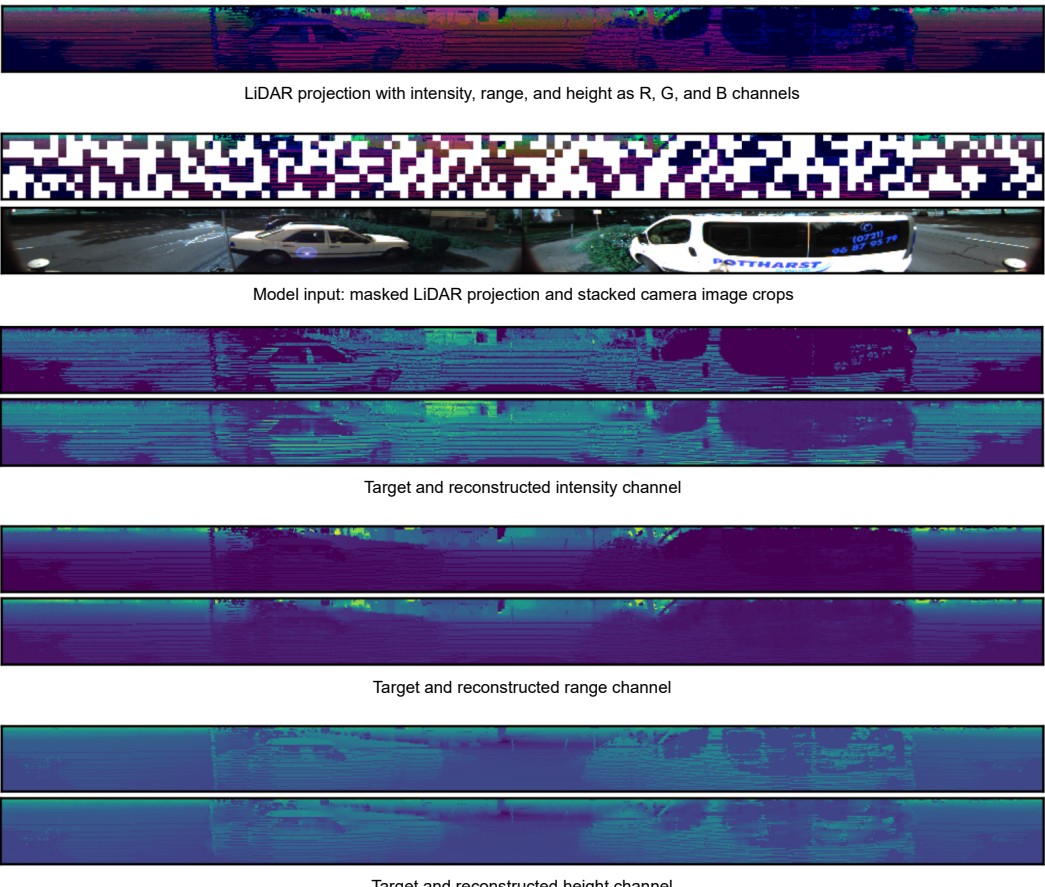

LiDAR projection with intensity, range, and height as R, G, and B channels

Model input: masked LiDAR projection and stacked camera image crops

Target and reconstructed intensity channel

Target and reconstructed range channel

Target and reconstructed height channel

Figure 4: **Validation sample and corresponding LiDAR reconstructions.** To improve the visibility, we pseudo-colorize all channels using the viridis colormap (Hunter, 2007).

