# OpenReview forum: "MaskedFusion360: Reconstruct LiDAR Data by Querying Camera Features"
_ICLR.cc/2023/TinyPapers — Submitted to Tiny Papers @ ICLR 2023_

### Official Review · Reviewer_27ym · 2023-03-20

**Confidence:** 4

**Summary Of Contributions:**

The paper proposes a MAE-inspired method for self-supervised representation learning with paired LiDAR and camera data. The LiDAR data is masked and masked areas are reconstructed via cross-attention with camera data.

**Rating:**

Great Start (GS): a submission which meets some of the reviewing criteria but has room for improvement

**Strengths And Weaknesses:**

Strengths
* The paper is clearly written and Fig 1 is very useful for giving an overview
* Good references to existing literature
* The cross-modal application of MAEs to LiDAR+camera data is an interesting direction to investigate as self-supervised learning could help to make perception models much more data efficient.

Weaknesses
* It would be best practice to keep a held-out test set - to help avoid overfitting on a validation set.
* It wasn't clear to me that range and intensity of the lidar were being reconstructed until Fig 3. It may be worth adding some detail about that?
* If camera features are held out (Fig 2), how can reconstruction be achieved at all? Fig 1 suggests there is no self-attention mechanism for LiDAR features - should this be added? In that case would that be like training a regular MAE on LiDAR-only data?



**Suggested Changes:**

I rated GS for the moment due to some of the items in the weaknesses section above which require clarification. If these can be reasonably addressed, I think the paper will be between HP<>CCR. To expand the work into a full paper in the future, it would be interesting to run a comparative study of vision-only vs lidar-only vs the cross-modal approach here, and to better understand the usefulness of the learned representations - e.g. using them to measure improvements in few-shot learning of 3D detection models. A closely related work from ECCV last year which might provide further inspiration is https://multimae.epfl.ch/

Minor suggestions / typos
* "lacks the semantic richness" - this term is a bit vague, maybe more specifically e.g. lacks the photometric qualities and sampling density of camera data?
* "Therefore, we reduce the learned transformations to moderate perspective transformation and do not require additional modules to generate dense LiDAR representations." - this
* up-on -> upon
* 16x16 -> $16\times16$, same for 8x8
* the main text says a patch size of 16 was used, the appendix says 8? Is one for LiDAR, the other for camera?
* querys -> queries
* two sentences are a bit repetitive and could be combined - "Accordingly, our model learns to reconstruct LiDAR data by querying camera features. Finally, a MAE decoder reconstructs the spherical LiDAR projections from the fused tokens."
* Fig 3: difference/error images may be interesting to plot
* Figs 2+3: add text for what each image row is to make comparison easier for reviewers
* why train for 57 epochs? What was the stopping criteria?

---

> ### Author Response · Authors · 2023-05-27
> **Comment on the paper revision**
>
> Thank you for the feedback.
> We have adjusted the paper to address the weaknesses and have taken into account the suggested changes.
> This is the list of corresponding changes:
> > Weakness: keep a held-out test set to help avoid overfitting on a validation set
> - Unfortunately, when we submitted the paper, the test split for the KITTI dataset did not yet contain fish-eye camera images. We contacted the team behind the dataset and they have now published the fisheye camera images. We have added the results of our models on the test set in Table 1 in Appendix B.
>
> > Weakness: It wasn't clear to me that range and intensity of the lidar were being reconstructed until Fig 3
> - We have added this information in Figure 1 and described it in Appendix B.
>
> > Weakness: If camera features are held out (Fig 2), how can reconstruction be achieved at all? Fig 1 suggests there is no self-attention mechanism for LiDAR features - should this be added?
> - Both encoders and the decoder are ViT-based, accordingly there are several self-attention mechnisms for LiDAR tokens even without camera features. In addition, the fusion block in this case acts as a gating mechanism for the LiDAR class token, making our model comparable to a pure LiDAR autoencoder. We have described this in detail in Appendix A and added the corresponding skip connection in Figure 1.
>
> > Suggestion: the main text says a patch size of 16 was used, the appendix says 8? Is one for LiDAR, the other for camera?
> * That's a misunderstanding. The patch size of 16 is used in a vanilla MAE, we use a patch size of 8 for both our encoders. We moved that into the Appendix B to clarify it.
>
> > Suggestion: Figs 2+3: add text for what each image row is to make comparison easier for reviewers
> * We added this information to Fig.3. Unfortunately, there is not enough space to add it to Fig.2 and still stick to the page limit of 2 pages of main text.
>
> > Suggestion: why train for 57 epochs? What was the stopping criteria?
> * We train for 57 epochs since the model converges doesn't improve further (early stopping, see Figure 2 right).
>
> In addition, we have made the code for our work publicly available to ensure reproducibility.

---

### Official Review · Reviewer_VZVT · 2023-04-01

**Confidence:** 5

**Summary Of Contributions:**

The submission presents a self-supervised method for fusing LiDAR and camera data in self-driving applications. The approach builds upon MAEs to reconstruct masked LiDAR data by fusing features from dense spherical LiDAR projections and features from fish-eye camera crops with a similar field of view.

**Rating:**

Great Start (GS): a submission which meets some of the reviewing criteria but has room for improvement

**Strengths And Weaknesses:**

### Strengths:
1. The submission is well-motivated and clear.
2. The proposed method is simple and requires less overhead compared to related works.

### Weaknesses:
- **Reproducibility** : Although the authors have used the open-source KITTI dataset, they have not provided the necessary code to reproduce the results. Furthermore, the training-validation split seems small (92%-8%), raising concerns about the generalizability of the approach to different random samplings of the dataset.
- **Correctness** :  While the authors provide qualitative results demonstrating the effectiveness of their approach, they have not conducted a quantitative evaluation of the method against at least one of the existing related works to determine its correctness.


**Suggested Changes:**

1. While I find the proposed idea simple yet promising, I suggest that the authors compare their approach with at least one baseline to provide a justification for their claims.
2. Additionally, in the "Introduction" section, the authors state that their "contributions are twofold...we perform evaluation on a self-driving dataset". However, performing evaluation on a dataset is not a contribution but rather a means of justifying the correctness of the approach. I recommend that the authors rephrase this sentence.
3. **Clarification**: In Section 2 "Method," the authors state that "our LiDAR encoder achieves the same computational efficiency as a vanilla MAE encoder." However, in the appendix, they mention that they used a smaller patch size (8x8) compared to the vanilla MAE. Therefore, the computational efficiency should be different. I recommend the authors provide further clarification in this regard.

---

> ### Author Response · Authors · 2023-05-27
> **Comment on the paper revision**
>
> Thank you for the feedback. We have adjusted the paper to address the weaknesses and have taken into account the suggested changes. This is the list of corresponding changes:
>
> > Weakness: Reproducibility and generalizability
> * We have made the code for our work publicly available and additionally evaluated our models on the test split of the KITTI-360 dataset. We added the results to Table 1 in Appendix B. (Unfortunately, when we submitted the paper, the test split for the KITTI-360 dataset did not yet contain fish-eye camera images. We contacted the team behind the dataset and they have now published the fisheye camera images.)
>
> > Weakness/ suggested change: Correctness/ compare to related works
> * If no camera features are used, the fusion block acts as a gating mechanism for the LiDAR class token and our model becomes comparable to a pure LiDAR masked autoencoder (MAE). We have described this in detail in Appendix A and listed the results of the two variants in Table 1.
>
> > Suggested change: performing evaluation on a dataset is not a contribution
> * We removed this sentence.
>
> > Clarification: Same computational efficiency with different patch sizes?
> * That's a misunderstanding. We achieve the same computational efficiency with respect to the fact that our encoder is also executed only on non-masked tokens. Masked tokens are re-introduced in the encoder-decoder projection after the encoder (Figure 1). We have moved the information on patch sizes to Appendix A to avoid this misunderstanding.

---

### Comment · Area_Chair_qxqW · 2023-06-04
**Revised version**

This work meets the threshold for archival, contains the URM statement and is deanonymized.

---

### Meta-Review · Area_Chair_qxqW · 2023-04-08

**Recommendation:** Invite to revise
**Confidence:** 4

**Metareview:**

A self-supervised representation learning for fusing LiDAR and camera data in self-driving applications.

The paper is written well, and including Fig 1 is welcomed. The literature was well-referenced and showed an understanding of the field.

The questions on correctness and reproducibility are raised due to the lack of code and the practices around the train-test set split and overfitting. How different is the proposed approach?


**Summary:**

A self-supervised representation learning for fusing LiDAR and camera data in self-driving applications.

**Comments And Feedback To The Authors:**

Read the reviews and make the suggested changes.

**Reason For Not Giving A Higher Recommendation:**

This paper is a great start and has the potential to be a good paper. However, in the current state, the authors need to clarify the differences and compare them with the existing methods. Also, authors need to consider reproducibility and provide open-source code. Please be careful with overfitting.

**Reason For Not Giving A Lower Recommendation:**

N/A

---

> ### Author Response · Authors · 2023-05-27
> **Comment on the paper revision**
>
> Thank you for the feedback. We have adjusted the paper to address the weaknesses and have taken into account the suggested changes (see comments below). Most notably, we additionally evaluated our models on the test split of the KITTI-360 dataset and published the code.
> Additionally, we changed the name of our method from MaskedFusion to MaskedFusion360, since MaskedFusion is already used in another unrelated paper (https://arxiv.org/abs/1911.07771).

---

### Decision · Program_Chairs · 2023-04-09

Revision accepted; invite to archive

---

> ### Author Response · Authors · 2023-05-31
> **Opt-in for archival**
>
> We wish to opt-in for archival.